# CGNF: Conditional Graph Neural Fields

## Abstract

Graph convolutional networks have achieved tremendous success in the tasks of graph node classification. These models could learn a better node representation through encoding the graph structure and node features. However, the correlation between the node labels are not considered. In this paper, we propose a novel architecture for graph node classification, named conditional graph neural fields (CGNF). By integrating the conditional random fields (CRF) in the graph convolutional networks, we explicitly model a joint probability of the entire set of node labels, thus taking advantage of neighborhood label information in the node label prediction task. Our model could have both the representation capacity of graph neural networks and the prediction power of CRFs. Experiments on several graph datasets demonstrate effectiveness of CGNF.

## 1 Introduction

Graph is an expressive representation for diverse datasets. Node classification on graphs has many real world applications. For example, to classify publications we build citation networks where publications are nodes and the research fields are node labels (Hamilton et al., 2017). To study the protein-disease relations we could construct protein-protein interaction (PPI) graphs where each protein is a node and diseases are treated as node labels. Node classification on graphs is a general task that creates knowledge and has real social impact, e.g., by exploring the similarities between proteins, we can learn novel protein target for treating diseases and use that knowledge to facilitate drug repurposing and safety monitoring (Zitnik et al., 2018; Ma et al., 2018).

Many earlier works take connectivity-based approaches to classify nodes, which mainly operate on the graph structure alone. The underlying assumption is that nodes that are closely connected in the graph should have similar labels. Among the most successful connectivity-based methods are node embedding techniques including (Perozzi et al., 2014; Grover & Leskovec, 2016a; Tang et al., 2015a; Hamilton et al., 2017; Kipf & Welling, 2016). More recently, features of neighboring nodes are included in learning node embedding. These feature-based approaches incorporate graph connectivities via regularization (Zhu et al., 2003; Belkin et al., 2006) or aggregation over local neighborhoods (Defferrard et al., 2016; Kipf & Welling, 2017; Chen et al., 2018a). Among them, the graph convolutional networks (GCN) become one of the most successful methods of node embedding for arbitrary graphs with node features (Kipf & Welling, 2017; Defferrard et al., 2016; Hamilton et al., 2017; Chen et al., 2018a).

Although GCN gets good node embeddings and node classification performance via aggregating node features and graph connectivities, the correlation of node labels was not exploited. In practice, the labels of neighborhood nodes in a graph are often mutually impacted. For example, in citation networks the labels (fields of research) for neighborhood nodes (publications) are often consistent: a paper citing another tends to appear in similar or related fields to the cited one. Unfortunately, we are not able to simply treat labels as additional node features because label information are not available in test set. Instead, the key question is *how to leverage label correlation in addition to node features in node classification*.

We notice that energy-based approaches such as the conditional random fields (CRF) (Lafferty et al., 2001) are desirable for capturing structured labels including correlation of node labels. However, they may involve non-trivial optimization to perform predictions.

In this paper we propose a new deep architecture to extend the GCN by defining an energy function to capture the dependencies between labels for better node embedding and classification. In particular,

we model the pairwise label relation with a pairwise energy function and propose a new model integrating graph node embedding and conditional random fields. The proposed model, coined conditional graph neural fields (CGNF), has the following contributions:

- CGNF is the first work that bridges GCN and CRF to directly capture label correlations for more accurate node classification in graphs. Experimental results on several graph datasets demonstrated the advantage of CGNF over several baselines including GCN.
- We designed a specific graph-based energy function and proposed an efficient way for approximate training as well as two efficient inference schemes for predicting test labels.
- Experimental results on several graph datasets demonstrate our advantage over other graph neural networks such as GCN.

## 2 BACKGROUND

**Graph Convolutional Networks** Over the past few years, several graph convolutional network models emerged to compute informative node feature representations for arbitrary graphs (Kipf & Welling, 2017; Defferrard et al., 2016; Hamilton et al., 2017; Chen et al., 2018a). GCN models learn node embeddings in the following manner: Given each graph node initially attached with a feature vector, the embedding vector of each node are the transformed weighted sum of the feature vectors of its neighbors. All nodes are simultaneously updated to perform a layer of forward propagation over neural networks. The deeper the network, the larger the local neighborhood. Thus global information is disseminated to each graph node for learning better node embeddings. Specifically, given an undirected graph with input features matrix of nodes $\boldsymbol{X}$ and adjacency matrix of the underlying graph $\boldsymbol{A}$, a multi-layer neural network is constructed on the graph with the following layer-wise propagation rule:

$$\boldsymbol{H}^{(l+1)} = \sigma\left(\tilde{\boldsymbol{D}}^{-\frac{1}{2}}\tilde{\boldsymbol{A}}\tilde{\boldsymbol{D}}^{-\frac{1}{2}}\boldsymbol{H}^{(l)}\boldsymbol{W}^{(l)}\right) \tag{1}$$

where $\tilde{\boldsymbol{A}} = \boldsymbol{A} + \boldsymbol{I}$ is the adjacency matrix with self-connections. $\boldsymbol{I}$ is the identity matrix, $\boldsymbol{D}$ is a diagonal matrix such that $\tilde{\boldsymbol{D}}_{ii} = \sum_j \tilde{\boldsymbol{A}}_{ij}$, $\boldsymbol{W}^{(l)}$ is a layer-specific parameter matrix, $\boldsymbol{H}^{(l)}$ is the node representation in the $l^{th}$ layer, $\boldsymbol{H}^{(0)} = \boldsymbol{X}$, and $\sigma$ is an activation function (e.g. ReLU or sigmoid). However, labels and label correlation is not modeled in GCN.

**Conditional Random Fields** The conditional random fields (CRF) is an undirected probabilistic graphical model commonly used for structured prediction tasks, e.g. named entity recognition and image segmentation. Formally, given input feature $\boldsymbol{x} \in \mathbb{R}^d$, CRF aims at finding the label set $\boldsymbol{y}$ that maximizes the conditional probability $P(\boldsymbol{y}|\boldsymbol{x})$. In particular, CRF defines a joint probability over cliques $\{\boldsymbol{x}_c\}$ in the graph. A clique is any fully connected subset of the the graph. CRF represents $P(\boldsymbol{y}|\boldsymbol{x})$ as a product of clique potential $\Phi_c(\boldsymbol{x}_c, \boldsymbol{y}_c)$. Each potential function $\Phi_c$ depends only on part of the feature input $\boldsymbol{x}$ denoted as $\boldsymbol{x}_c$ and a label subset $\boldsymbol{y}_a$. This factorization can allow us to represent $P(\boldsymbol{y}|\boldsymbol{x})$ much more efficiently (Sutton et al., 2012). By making sure the factorization form of $P(\boldsymbol{y}|\boldsymbol{x})$ sum to 1, partition function $Z(\boldsymbol{x})$ is used for normalization use as follows:

$$P(\boldsymbol{y}|\boldsymbol{x}) = \frac{1}{Z(\boldsymbol{x})} \prod_c \Phi_a(\boldsymbol{x}_c, \boldsymbol{y}_c) \tag{2}$$

where $Z(\boldsymbol{x}) = \sum_{\boldsymbol{y}'_c} \prod_c \Phi_a(\boldsymbol{x}_c, \boldsymbol{y}'_c)$ is a normalizer required for a valid probability distribution.

## 3 CGNF METHOD

### 3.1 TASK FORMULATION

Consider the problem of multi-class node classification. The input graph $G = \{\boldsymbol{X}, \boldsymbol{Y}, \boldsymbol{A}\}$ includes node feature matrix $\boldsymbol{X}$, adjacency matrix $\boldsymbol{A}$, and the output node label matrix $\boldsymbol{Y}$. More specifically, we have $\boldsymbol{X} = [\boldsymbol{x}_1, \boldsymbol{x}_2, \cdots, \boldsymbol{x}_N]^\mathsf{T} \in \mathbb{R}^{N \times D}$ and $\boldsymbol{Y} = [\boldsymbol{y}_1, \boldsymbol{y}_2, \cdots, \boldsymbol{y}_N]^\mathsf{T} \in \mathbb{R}^{N \times M}$, where $N$ is the number of nodes in the graph, $D$ is the dimension of node features, and $M$ is the number of distinct labels. The $i^{th}$ node has a feature vector $\boldsymbol{x}_i$, and a one-hot label vector $\boldsymbol{y}_i$. The graph connectivity is specified through the adjacency matrix $\boldsymbol{A} \in \mathbb{R}^{N \times N}$, where $A_{i,j} \in \boldsymbol{A}$ indicates the edge weight

Table 1: Notation references

| Notation | Definition |
|---|---|
| $G : \{\boldsymbol{X}, \boldsymbol{Y}, \boldsymbol{A}\}$ | the whole graph data |
| $\boldsymbol{X} \in \mathbb{R}^{N \times D}$ | node feature matrix |
| $\mathcal{Y}$ | label space consisting of unique label assignments |
| $\boldsymbol{Y} \in \mathbb{R}^{N \times M}$ | label vectors for all nodes |
| $\boldsymbol{A} \in \mathbb{R}^{N \times N}; \hat{\boldsymbol{A}} \in \mathbb{R}^{N \times N}$ | graph adjacency matrix; a normalized version of $\boldsymbol{A}$ |
| $f(\cdot)$ | graph convolutional networks (GCN) |
| $\boldsymbol{H} \in \mathbb{R}^{N \times M}; \boldsymbol{h}_i \in \mathbb{R}^M$ | GCN prediction probability distribution; row $i$ of $\boldsymbol{H}$ |
| $\boldsymbol{W}^0 \in \mathbb{R}^{D \times S}, \boldsymbol{W}^1 \in \mathbb{R}^{S \times M}$ | learnable weight matrix of GCN |
| $Z(\cdot)$ | partition function |
| $N(i)$ | neighbor node set of node $i$ |
| $E(\cdot)$ | energy function |
| $\psi(\cdot); \phi(\cdot)$ | unary potential of $E$; pairwise potential of $E$ |
| $\boldsymbol{U} \in \mathbb{R}^{M \times M}$ | learnable correlation weight matrix |

between node $i$ and node $j$. We follow convention to denote $\hat{\boldsymbol{A}} = \tilde{\boldsymbol{D}}^{-\frac{1}{2}} \tilde{\boldsymbol{A}} \tilde{\boldsymbol{D}}^{-\frac{1}{2}}$ as a normalized version of $\tilde{\boldsymbol{A}}$, which will be used in our model later.

We follow the traditional setting of GCN and formalize the learning in semi-supervised setting where we assume we observe the labels $\boldsymbol{Y}_{tr}$ only for some of the nodes $\boldsymbol{X}_{tr}$. Our goal is to predict the labels $\hat{\boldsymbol{Y}}_{te}$ for other unlabeled nodes $\boldsymbol{X}_{te}$. To start with, we consider $\{\boldsymbol{X}_{tr}, \boldsymbol{Y}_{tr}, \boldsymbol{A}_{tr}\}$ as training set, and $\{\boldsymbol{X}_{te}, \boldsymbol{Y}_{te}, \boldsymbol{A}_{te}\}$ as test set.

## 3.2 THE CGNF MODEL

Given input graph $G = \{\boldsymbol{X}, \boldsymbol{Y}, \boldsymbol{A}\}$ as previous described, our goal is to predict node labels $\boldsymbol{Y}$ for graph $G$. We can learn node embeddings and use them to make such predictions. In this paper, we follow (Kipf & Welling, 2017) and use a two-layer GCN model given by Eq. 3.

$$\boldsymbol{H} = f(\boldsymbol{X}, \boldsymbol{A}) = \text{Softmax}(\hat{\boldsymbol{A}} \text{ReLu}(\hat{\boldsymbol{A}} \boldsymbol{X} \boldsymbol{W}^0) \boldsymbol{W}^1) \tag{3}$$

Here $\boldsymbol{W}^0 \in \mathbb{R}^{D \times S}, \boldsymbol{W}^1 \in \mathbb{R}^{S \times M}$ are weight parameters of GCN. Softmax function is applied row-wise to make each row $\{\boldsymbol{h}_i\}_1^N$ of GCN prediction $\boldsymbol{H} \in \mathbb{R}^{N \times M}$. However, the training of GCN is conditionally independent for each node, and label correlation is ignored by this model.

To fill this gap, we aim to integrate label correlations into the GCN model. Energy based models, such as CRF, are flexible ways to define the variable dependencies. So in a general graph, considering both the impact of node features and label dependency, we can define the energy function as belows.

$$E(\boldsymbol{Y}, \boldsymbol{X}, \boldsymbol{A}) = E_c(\boldsymbol{Y}_c, \boldsymbol{X}_c, \boldsymbol{A}) = \sum_i \psi(\boldsymbol{y}_i, \boldsymbol{x}_i) + \gamma \sum_{(i,j) \in \mathcal{E}, i < j} \phi(\boldsymbol{y}_i, \boldsymbol{y}_j, A_{i,j}) \tag{4}$$

where $c$ is a clique, $\mathcal{E} \in \mathbb{R}^M$ is the edge set, the unary potential $\psi(\cdot)$ is designed to measure the compatibility between observed node $\boldsymbol{x}_i$ and its label $\boldsymbol{y}_i$; and pairwise potential $\phi(\cdot)$ is meant for capturing label correlation; $\gamma$ is a mixture weight. With this energy function, we can derive the Gibbs distribution, i.e. the conditional probability $P(\boldsymbol{Y}|\boldsymbol{X}, \boldsymbol{A})$ as given by Eq. 5.

$$P(\boldsymbol{Y}|\boldsymbol{X}, \boldsymbol{A}) = \frac{\exp(-E(\boldsymbol{Y}, \boldsymbol{X}, \boldsymbol{A}))}{\sum_{\boldsymbol{Y}' \in \mathcal{Y}} \exp(-E(\boldsymbol{Y}', \boldsymbol{X}, \boldsymbol{A}))} = \frac{\exp(-E(\boldsymbol{Y}, \boldsymbol{X}, \boldsymbol{A}))}{Z(\boldsymbol{X}, \boldsymbol{A})} \tag{5}$$

where $Z = \sum_{\boldsymbol{Y}' \in \mathcal{Y}} \exp(-E(\boldsymbol{Y}', \boldsymbol{X}, \boldsymbol{A}))$ is the normalizer (or called partition function), $E$ is the energy function which is an indicator of the likelihood of the corresponding relationships within the clique, with a higher energy configuration having lower probability; $\mathcal{Y}$ is the label space consisting of unique label assignments $\boldsymbol{Y}'$. Then our objective is to maximize this conditional probability.

To enhance Eq. 4 by the neural network representations of GCN, the energy function can be reformulated as:

$$E(\boldsymbol{Y}, \boldsymbol{X}, \boldsymbol{A}) = \sum_i \psi(\boldsymbol{y}_i, \boldsymbol{h_i}) + \gamma \sum_{(i,j)\in\mathcal{E}, i<j} \phi(\boldsymbol{y}_i, \boldsymbol{y}_j, \hat{A}_{i,j}) \tag{6}$$

where $\boldsymbol{h_i}$ is the previously mentioned GCN prediction $\boldsymbol{H}$ on node $i$ and $\hat{A}_{i,j}$ is an element in the normalized adjacency matrix $\hat{\boldsymbol{A}}$. Separating the energies for each node, we can further express the energy function as Eq. 7.

$$E(\boldsymbol{Y}, \boldsymbol{X}, \boldsymbol{A}) = \sum_i \left( \psi(\boldsymbol{y}_i, \boldsymbol{h_i}) + \frac{\gamma}{2} \sum_{j\in N(i)} \phi(\boldsymbol{y}_i, \boldsymbol{y}_j, \hat{A}_{i,j}) \right) \tag{7}$$

where $N(i)$ is the neighborhood of node $i$.

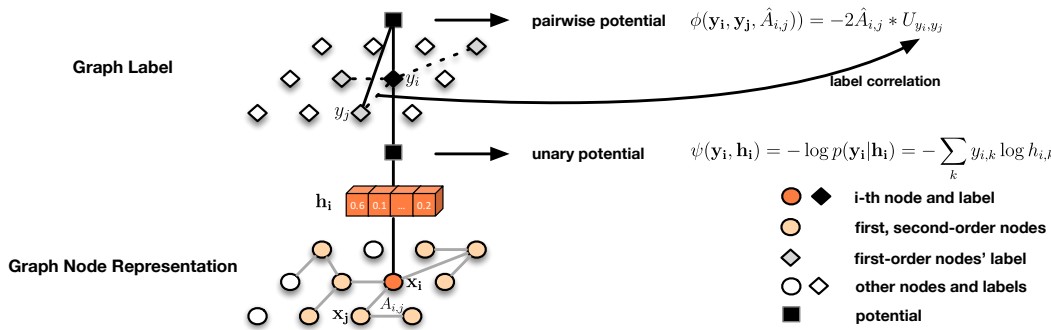

Figure 1: Graphical structure of CGNF. Here, we focus on node $i$. Initially, node $i$ in graph has its input feature $\boldsymbol{x}_i$ and is connected to other node such as node $j$ with edge weight denoted as $A_{i,j}$. First and second order neighbor nodes of node $i$ are utilized in two-layer GCN setting to generate graph prediction probability $\boldsymbol{h_i} \in \boldsymbol{H}$ using Eq. 3 and at the meantime we can get normalized graph adjacency matrix $\hat{\boldsymbol{A}}$. Then we utilize specific designed graph-based energy function in Eq. 7 consisting of unary and pairwise potential in Eq. 8 to make accurate node classification using Eq. 13.

Specifically, in a graph we define the unary potential for node $i$ as the prediction loss of GCN, denoted as $\psi(\boldsymbol{y}_i, \boldsymbol{h}_i)$. We also define the pairwise potential between node $i$ and node $j$ as $\phi(\boldsymbol{y}_i, \boldsymbol{y}_j, \hat{A}_{i,j})$, which is impacted by two factors: the normalized edge weights $\hat{A}_{i,j}$ and label correlation weights $U_{y_i,y_j}$. The two potential functions are computed as Eq. 8.

$$\psi(\boldsymbol{y}_i, \boldsymbol{h}_i) = -\log p(\boldsymbol{y}_i|\boldsymbol{h}_i) = -\sum_k y_{i,k} \log h_{i,k} \tag{8}$$

$$\phi(\boldsymbol{y}_i, \boldsymbol{y}_j, \hat{A}_{i,j}) = -2\hat{A}_{i,j} U_{y_i,y_j}$$

where $y_i$ and $y_j$ are the corresponding label index of label vector $\boldsymbol{y}_i$ and $\boldsymbol{y}_j$; and $U_{y_i,y_j} \in \boldsymbol{U}$ is the learnable correlation/transition weight between label $y_i$ and $y_j$.

Like in the traditional CRF, when all components for conditional probability in Eq. 5 are introduced throughly above, a common way is to use negative log likelihood as training objective function in Eq. 9 to find optimal parameters like $\boldsymbol{W}^0, \boldsymbol{W}^1, \boldsymbol{U}$:

$$\begin{aligned} -\log P(\boldsymbol{Y}|\boldsymbol{X}, \boldsymbol{A}) &= E(\boldsymbol{Y}, \boldsymbol{X}, \boldsymbol{A}) + \log Z(\boldsymbol{X}, \boldsymbol{A}) \\ &= E(\boldsymbol{Y}, \boldsymbol{X}, \boldsymbol{A}) + \log \sum_{\boldsymbol{Y}'} \exp(-E(\boldsymbol{Y}', \boldsymbol{X}, \boldsymbol{A})) \end{aligned} \tag{9}$$

Using the energy based conditional probability, the inference of new labels is comparably easy in this model. Notice that in Eq. 9 Since $P(\boldsymbol{Y}|\boldsymbol{X}, \boldsymbol{A}) \propto \exp(-E(\boldsymbol{Y}, \boldsymbol{X}, \boldsymbol{A}))$, after getting the parameters in the training phase, the inference of the test labels will simply be $\min_{\boldsymbol{Y}} E(\boldsymbol{Y}, \boldsymbol{X}, \boldsymbol{A})$. However,

how to optimize the parameters by minimizing the negative log likelihood is a great challenge. The computation of the normalizer $Z(\boldsymbol{X}, \boldsymbol{A})$ is often intractable, which causes a big problem in training. In the next section, we will introduce two ways to solve the problem.

## 3.3 PARAMETER OPTIMIZATION

In this section, we introduce the training ( parameter optimization) for CGNF. It is implemented on the training data $\boldsymbol{Y}_{tr}, \boldsymbol{X}_{tr}, \boldsymbol{A}_{tr}$ , but for simplicity here we omit the subscripts for all $\boldsymbol{Y}, \boldsymbol{X}, \boldsymbol{A}$ .

As we know, exact minimization of the energy based function Eq. 9 is intractable. So we consider a simple but effective approximation method, the pseudo likelihood method. A pseudo likelihood (Besag, 1975) provides a good approximation of the joint probability. Use of the pseudo likelihood (PL) in place of the true likelihood function in a maximum likelihood analysis can lead to good estimates.

$$P(\boldsymbol{Y}|\boldsymbol{X}, \boldsymbol{A}) \approx PL(\boldsymbol{Y}|\boldsymbol{X}, \boldsymbol{A}) = \prod_i P(\boldsymbol{y}_i|\boldsymbol{y}_{N(i)}, \boldsymbol{X}, \boldsymbol{A}) \tag{10}$$

where $N(i)$ are the neighbors of node $\boldsymbol{x}_i$. The pseudo likelihood has the significant advantage that it only requires normalizing over the possible labels at one node. It could reduce the computation complexity of the normalizer from $O(M^N)$ to $O(MN)$.

Recall the conditional probability given by Eq 5 and Eq 7, for each node $\boldsymbol{x_i}$, the conditional probability given its neighborhood is calculated as follows:

$$\begin{aligned} P(\boldsymbol{y}_i|\boldsymbol{y}_{N(i)}, \boldsymbol{X}, \boldsymbol{A}) &= \frac{\exp(-\psi(\boldsymbol{y}_i, \boldsymbol{h_i}) - \gamma \sum_{j \in N(i)} \phi(\boldsymbol{y}_i, \boldsymbol{y}_j, \hat{A}_{i,j})}{\sum_{\boldsymbol{y}_i'} (\exp(-\psi(\boldsymbol{y}_i', \boldsymbol{h_i}) - \gamma \sum_{j \in N(i)} \phi(\boldsymbol{y}_i', \boldsymbol{y}_j, \hat{A}_{i,j}))} \\ &= \frac{\exp(-\log p(\boldsymbol{y}_i|\boldsymbol{h_i}) - 2\gamma \sum_{j \in N(i)} \hat{A}_{i,j} U_{y_i, y_j}}{\sum_{\boldsymbol{y}_i'} (\exp(-\log p(\boldsymbol{y}_i'|\boldsymbol{h_i}) - 2\gamma \sum_{j \in N(i)} \hat{A}_{i,j} U_{y_i', y_j})} \end{aligned} \tag{11}$$

$\boldsymbol{y}_i'$ is all the possible label for node $\boldsymbol{x}_i$. Notice here we do not have the $\frac{1}{2}$ coefficient on the pairwise potential, as we need to extract all dependent potentials of node $\boldsymbol{x}_i$.

The new training objective, i.e. the negative log pseudo-log-likelihood, becomes the following form:

$$-\log PL(\boldsymbol{Y}|\boldsymbol{X}, \boldsymbol{A}) = \sum_i -\log P(\boldsymbol{y}_i|\boldsymbol{y}_{N(i)}, \boldsymbol{X}, \boldsymbol{A}) = \tag{12}$$

$$\sum_i \left( \psi(\boldsymbol{y}_i, \boldsymbol{h_i}) + \gamma \sum_{j \in N(i)} \phi(\boldsymbol{y}_i, \boldsymbol{y}_j, \hat{A}_{i,j}) + \log \sum_{\boldsymbol{y}_i'} (\exp(-\psi(\boldsymbol{y}_i', \boldsymbol{h_i}) - \gamma \sum_{j \in N(i)} \phi(\boldsymbol{y}_i', \boldsymbol{y}_j, \hat{A}_{i,j})) \right)$$

$$= -\sum_{i,k} (\boldsymbol{Y} \odot \log \boldsymbol{H})_{i,k} - 2\gamma \sum_{i,j,i \neq j} (\hat{\boldsymbol{A}} \odot (\boldsymbol{Y}\boldsymbol{U}\boldsymbol{Y}^T))_{i,j} + \sum_i \log \sum_k (\boldsymbol{H} \odot \exp(2\gamma\hat{\boldsymbol{A}}\boldsymbol{Y}\boldsymbol{U}))_{i,k}$$

where $i$ is the index of training nodes; $(\cdot)_{i,j}$ is the element of a matrix at row $i$ and column $j$; and $\odot$ indicates element-wise multiplication.

## 3.4 INFERENCE OF TEST LABELS FOR CGNF

In the previous section, we introduced how to train the model and get the optimized parameter estimation. As we explained in 3.2, after getting the model parameters, next the inference for new labels can be performed by optimizing Eq. 13.

$$\min_{\hat{\boldsymbol{Y}}_{te}} E(\hat{\boldsymbol{Y}}_{te}, \boldsymbol{X}, \boldsymbol{A}, \boldsymbol{Y}_{tr}) = \min_{\hat{\boldsymbol{Y}}_{te}} \left[ -\log p(\hat{\boldsymbol{Y}}_{te}|\boldsymbol{H}) - \gamma \sum_{i \neq j} (\hat{\boldsymbol{A}} \odot (\hat{\boldsymbol{Y}}\boldsymbol{U}\hat{\boldsymbol{Y}}^T))_{i,j} \right] \tag{13}$$

where $\hat{\boldsymbol{Y}} = \text{concatenate}(\boldsymbol{Y}_{tr}, \hat{\boldsymbol{Y}}_{te})$. To find the minimum value we have to find the "best" test labels, which is exponentially complex. Hence we approximate the inference in two ways. The simplest

way is not to consider the correlation between test labels just use training labels to infer one test label each time.

$$y_i = \arg\min_{y_j} E(\boldsymbol{y}_i, \boldsymbol{Y}_{tr}, \boldsymbol{X}, \boldsymbol{A}) = \arg\min_j [-\log(\boldsymbol{h}_i) - 2\gamma \hat{\boldsymbol{A}}_{tr} \boldsymbol{Y} \boldsymbol{U}^T]_j \qquad (14)$$

Alternatively, we could use a dynamic-programming-style (DP) method to find the minimum. We randomly select a test node as the start and randomly permute the other test nodes. Then we can do beam search along the order of the permuted test nodes (each beam search with beam size $K$ can get $K$ best sets and we select just the best one). Let us repeat this process $T$ times, and then we compare all the $T$ beam-search results and select the best one. Then we can just select one best according to their energy. The inference time in this case will be linear. The algorithm is summarized below.

---

**Algorithm 1** Inference `CGNF` with dynamic programming

---

**Input:** Node feature matrix $\boldsymbol{X}$, adjacency matrix $\boldsymbol{A}$, times $T$ and beam size $K$, energy function $E$;
**Output:** Estimated best output $\hat{\boldsymbol{Y}}^*$;
  Initialize process times $t \leftarrow 0$; energy function score $s \leftarrow \inf$; candidate output $\boldsymbol{Y}_c \leftarrow \emptyset$
  **while** $t \leq T$ **do**
    Reset beam $\mathcal{B} \leftarrow \emptyset$;
    Randomly permute test nodes $\{i\}_1^{|Y_{tr}|}$ as $\{z_i\}_1^{|Y_{tr}|}$ where $z_i$ is the new index for node $i$;
    **for** $i = 1$ to $|Y_{te}|$ **do**
      **for all** $\hat{\boldsymbol{Y}}_{1:i} \in \mathcal{B}$ **do**
        $\{\hat{\boldsymbol{Y}}_{1:i}\} \leftarrow$ extend $\{\hat{\boldsymbol{Y}}_{1:i-1}\}$ with $y_i \in$ label set $Y$;
        Append topK($\{\hat{\boldsymbol{Y}}_{1:i}\}$) using $E$ into $\mathcal{B}$;
      **end for**
      $\mathcal{B} \leftarrow topK(\mathcal{B})$;
    **end for**
    $t \leftarrow t + 1$;
    Compute current minimum energy score $s' \leftarrow \min_{\hat{\boldsymbol{Y}} \in \mathcal{B}} E(\boldsymbol{X}, \boldsymbol{A}, \boldsymbol{Y}_{tr}, \hat{\boldsymbol{Y}})$;
    Compute current best output $\boldsymbol{Y}_c^* \leftarrow \arg\min_{\hat{\boldsymbol{Y}} \in \mathcal{B}} E(\boldsymbol{X}, \boldsymbol{A}, \boldsymbol{Y}_{tr}, \hat{\boldsymbol{Y}})$;
    **if** $s' \leq s$ **then**
      $s \leftarrow s'$;
      $\boldsymbol{Y}_c \leftarrow \boldsymbol{Y}_c^*$;
    **end if**
  **end while**
  **return** $\hat{\boldsymbol{Y}}^* \leftarrow \boldsymbol{Y}_c$;

---

### 3.5 INDUCTIVE LEARNING ACROSS GRAPHS

Following the framework of GCN, our node classification task is essentially a semi-supervised learning task in one fixed graph. However, sometimes we have several graphs which do have connection with each other. In this case we can conduct an inductive learning for `CGNF`. Compared with semi-supervised setting, the only difference of inductive learning is that we cannot see the test nodes in the training phase. Thus we need to predict $\boldsymbol{H}_{tr}$ only based on the training nodes $\boldsymbol{X}_{tr}$ and their adjacency matrix $\boldsymbol{A}_{tr}$. Then we can still use Eq. 13 for training.

For inference of the test labels, as we lose the connections between training nodes and test nodes, so the trainings labels cannot directly impact the inference of test labels. In fact, we can still form a graph containing both training data and test data, however $A_{i,j} = 0$ for training node $i$ and test node $j$. So the pairwise potential between training nodes and test nodes will also become 0 according to Eq. 8. So Eq. 14 is not useful anymore, because the pairwise potential becomes 0 if we just utilize the training labels for inference. Instead, in this case we will use the DP-style inference.

## 4 EXPERIMENTS

**Data** We evaluated the effectiveness of `CGNF` on the following benchmark tasks: (1) classifying research topics using the Cora citation dataset (McCallum et al., 2000); (2) categorizing academic

papers with the Pubmed database; (3) classify research areas using the Citeseer citation dataset (Sen et al., 2008); and (4) classifying protein functions across various biological protein-protein interaction (PPI) graph (Borgwardt et al., 2005). Implementation details are in A.3 in Appendix.

Table 2: Dataset Statistics

| Dataset | Nodes | Edges | Classes | Features | Training/Validation/Test |
|---------|-------|-------|---------|----------|--------------------------|
| Cora | $2,708$ | $5,429$ | 7 | $1,433$ | $140/500/1000$ |
| Pubmed | $19,717$ | $44,338$ | 3 | $500$ | $60/500/1000$ |
| Citeseer | $3,327$ | $4,732$ | 6 | $3,703$ | $120/500/1000$ |
| PPI | $43,471$ | $81,044$ | 3 | $50$ | $120/500/1000$ |

**Baselines** To evaluate the effectiveness of our method, we compare with the following baselines.

- GCN (Kipf & Welling, 2017) learns latent representations by encoding graph structure and node features. It demonstrated significant improvement in semi-supervised node classification tasks.

- GraphSAGE (Hamilton et al., 2017) is an inductive variant of GCN which can generate node embedding for unseen data by sampling and aggregating features from a node's local neighborhood.

- FastGCN (Chen et al., 2018a) enhanced GCN with inductive ability and efficiency by interpreting graph convolutions as integral transforms of embedding function under probability measures. Monte Carlo approaches were used to estimate integrals and lead to batched training scheme.

- DeepWalk (Perozzi et al., 2014) learns node embedding by extracting node sequence from graph using truncated random walks. Skip-gram method is also utilized to maximum the conditional probability of neighbor nodes given context node. It is a state-of-the-art graph embedding method that showed effectiveness on large-scale social networks classification datasets.

- LINE (Tang et al., 2015b) is the follow-up method of DeepWalk. It considers both first-order proximity and second-order proximity by maximizing the joint probability between nodes.

- Node2Vec (Grover & Leskovec, 2016b) is a variant of DeepWalk by introducing two additional hyper-parameters $q, p$ to control transition probability of random walk. They approximate best first search and depth first search behavior for learning more informative representations.

For semi-supervised learning, we used the same setting for all the baselines, i.e. training the node embedding using all node features from both training data and test data. While for inductive learning, we use node features only from training data.

## 4.1 RESULTS

As shown in Table 3, `CGNF` is compared to a variety of high-performing baselines on a selection of benchmark graph node classification tasks. Results show `CGNF` models achieve the highest accuracy among all baselines with respect for all semi-supervised scenarios.

For skip-gram based methods including DeepWalk, Node2Vec and LINE, they perform worse than GCN-based models due to these methods are all inherently unsupervised learning methods with post-processing supervised classifiers. As a contrast, GCN-based methods combine semi- or supervised classification which can embed richer graph information to yield better performance. Notice that since GraphSAGE learns inductively, on Cora, Pubmed, and Citeseer it yields slightly lower accuracy than GCN.

Among these GCN-based methods, GraphSAGE is designed for inductive learning, thus performs best in the inductive task. However, under semi-supervised setting (e.g., on Cora, Pubmed and Citeseer datasets), it received very low accuracy due to having fewer labeled data.

Moreover, in inductive setting, skip-gram based methods require expensive additional training to inference on unseen test data which are inherently transductive. `CGNF` outperforms GCN and Fast-GCN since label correlations are useful in bridging training data and unseen graphs.

Table 3: Performance Comparison(micro-F1).

| Methods | Cora | Pubmed | Citeseer | PPI (inductive) |
|---|---|---|---|---|
| GCN | .813 | .790 | .709 | .488 |
| GraphSAGE | .766 | .779 | .657 | **.524** |
| FastGCN | .778 | .778 | .680 | .481 |
| DeepWalk | .753 | .723 | .479 | NA |
| Node2Vec | .743 | .727 | .517 | NA |
| LINE | .620 | .619 | .355 | NA |
| CGNF _PL(no DP) | **.832** | .792 | **.722** | NA |
| CGNF _PL(DP) | .823 | **.794** | .716 | .491 |

## 5 RELATED WORKS

The connectivity-based node classification approaches operate on the graph structure alone. Among the most successful connectivity-based methods are node embedding techniques including (Perozzi et al., 2014; Grover & Leskovec, 2016a; Tang et al., 2015a; Hamilton et al., 2017; Kipf & Welling, 2016). An underlying assumption of these techniques is that nodes which are closely connected in the graph, should have similar labels. In the feature-based setting the graph structure can be incorporated in different ways, for instance by using regularization (Zhu et al., 2003; Belkin et al., 2006), combining attributes with node embeddings, or aggregating them over local neighborhoods (Defferrard et al., 2016; Kipf & Welling, 2017; Chen et al., 2018a).

Incorporating probabilistic graphical model like conditional random fields (CRF) and neural networks is one possible approach to tackle structured prediction task in different domains. Initial works such as CNF (Peng et al., 2009) and NeuroCRF (Do & Artieres, 2010) yielded powerful models by adding neural networks layer after CRF's input. Similar idea shown in (Chen et al., 2018b) which applied CRF inference as a post-processing step after the training of deep convolutional neural networks for image segmentation task. Another line of works tried to integrate CRF with deep neural networks in end-to-end form see (Zheng et al., 2015; Ma & Hovy, 2016; Long et al., 2018; Vemulapalli et al., 2016). For example, (Zheng et al., 2015) formulated CRF with Gaussian pairwise potentials and mean-field approximate inference (Krähenbühl & Koltun, 2011) in recurrent neural networks way in image segmentation task, and for sequence labeling tasks, (Ma & Hovy, 2016) used a sequential CRF on top of DNNs to jointly decode labels for the whole sentence.

In addition, in Appendix, we also built a connection between CGNF and the graph based semi-supervised models.

## 6 CONCLUSION

In this paper we propose CGNF that extends GCN by defining an energy function to capture the dependencies between labels for better node embedding and classification. We proposed efficient ways to make the learning of CGNF tractable. Compared with GCN, the proposed model gained significant performance increase. Future direction includes exploring other efficient approximation method for training (such as mean-field inference) and applying this energy based framework to other graph neural networks.

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

# A    APPENDIX

## A.1    CONNECTION TO GRAPH-BASED SEMI-SUPERVISED MODEL

In some graph based semi-supervised learning models, the dependency between labels are also considered. In this section, we will build some connection between CGNF and these graph-based semi-supervised learning models. We will show some graph regularization term can be transformed to one special form of our pairwise potential. Take Zhu et al. (2003) as an example. They proposed the harmonic energy minimization method by minimizing the quadratic energy function Eq. 15.

$$E \propto \sum_i (f_i - y_i)^2 + \frac{1}{2} \sum_{i,j} w_{i,j}(f_i - f_j)^2 \tag{15}$$

Comparing to (13), we use $p(y_i|h_i)$ to replace the $\mathcal{L}_2$ loss, and use $U_{y_i,y_j}$ to take place of $(f_i - f_j)^2$; $w_{i,j}$ corresponds to the weight $\hat{A}_{i,j}$. $U_{y_i,y_j}$ is an unknown parameter, so we have to consider the normalizer in the training phase if we use negative log likelihood as the objective function. However, if we fix $U = I$, then the sum of the pairwise potential will become:

$$-\sum_i \sum_{j \in N(i)} \hat{A}_{i,j} U_{y_i,y_j} = -\sum_i \sum_{j \in N(i)} \hat{A}_{i,j} \zeta(y_i = y_j) = -\sum_i \sum_j \frac{1}{2} \hat{A}_{i,j}(2 - ||\boldsymbol{y_i} - \boldsymbol{y_j}||_2^2) \tag{16}$$

where $\zeta(y_i = y_j) = 1$ if $y_i = y_j$, and $\zeta(y_i = y_j) = 0$ if $y_i \neq y_j$; $||.||_2^2$ is the square of the $\mathcal{L}^2$ norm. As $\hat{A}$ is a constant, optimizing this energy will be identical to the Eq. 15.

In addition, the graph based semi-supervised model only considers label consistency. However, by defining correlation matrix $\boldsymbol{U}$, the label correlation integrated in our CGNF is much more flexible.

## A.2    IMPLEMENTATION DETAILS

For Cora, Pubmed, and Citeseer, we follow the experiment setup in (Kipf & Welling, 2017) to demonstrate the effective use of CGNF. For PPI, we consider the node classification task in inductive semi-supervised setting. Thus, PPI (inductive) is conducted where nodes in validation and test dataset are totally unseen. All methods are trained in training data; hyperparameters (e.g. dropout rate for all layers, number of hidden units) are only optimized on Cora based on the performance on validation dataset and are used for the rest datasets. We report the micro average F1 measure which is the same as accuracy on multiclass tasks.

We showed the specific dataset split ratio in Table. 2 which keeps the labels for training data in a small portion to test the methods in semi-supervised setting. For DeepWalk, Node2Vec, the number of random walks to start at each node is set to 10 and window size of skip-gram model is set to 10 by default. We choose the 2nd-order relation for LINE and save the embeddings of the best validation accuracy during training. One-vs-Rest technique is used based on logistic regression implemented using scikit-learn (Pedregosa et al., 2011) for DeepWalk, LINE and Node2Vec.

We train GCN-based methods for 200 epochs using Adam (Kingma & Ba, 2014) with a learning rate of .01 and early stopping with a window size of 30, i.e., we stop training if the current performance in validation set is lower than average of 30 consecutive epochs. They are implemented with two layers initialized using Glorot method (Glorot & Bengio, 2010) and appended with dropout (Srivastava et al., 2014) for each layer.

We report mean performance of 10 runs with random weight initializations for GCN-based methods. In detail, we set .5 dropout rate, 100 hidden size, $5 * 10^{-4}$ L2 regularization, 1 mixture weight $\gamma$ for CGNF and its variants. We compare against other GCN-based methods where we choose their best performing model or hyper-parameters setting as claimed in their paper.

