# OpenReview forum: "CGNF: Conditional Graph Neural Fields"
_ICLR.cc/2019/Conference_

### Official Review · AnonReviewer2 · 2018-11-01
**Reviewer comment**

**Rating:** 5
**Confidence:** 5

**Review:**

This paper proposed a combination of graph neural networks and conditional random field to model the correlation between node labels in the output.  In typical graph neural nets the predictions for nodes are conditionally independent given the node representations.  This paper proposes to use a CRF to compensate for that.  In terms of the approach, the authors used GCNs to produce unary potentials for the CRF, and have the pairwise potentials on each edge to model the correlation of labels of neighboring nodes.  Learning is done by optimizing pseudo-likelihood and the energy loss, while inference is performed through a couple heuristic processes.

Combining neural nets with CRFs is not a new idea, in particular this has been tried before on image and sequence CRFs.  It is therefore not surprising to see an attempt to also try it for graph predictions.  The main argument for using a CRF is its ability to model the correlations of output labels which was typically treated as independent.  However this is not the case for deep neural networks, as it already fuses information from all over the input, and therefore for most prediction problems it is fine to be conditionally independent for the output, as the dependence is already modeled in the representations.  This is true for graph neural networks as well, if we have a deep graph neural net, then the GNN itself will take care of most of the dependencies between nodes and produce node representations that are suitable for conditionally independent output predictions.  Therefore I’m not convinced that CRFs are really necessary for solving the prediction tasks tried in this paper.

The learning and inference algorithms proposed in this paper are also not very convincing.  CRFs has been studied for a long time, and there are many mature algorithms for learning them.  We could do proper maximum conditional likelihood learning, and use belief propagation to estimate the marginals to compute the gradients.  Zheng et al. (2015) did this for convnets, we could also do this for graph CRFs as belief propagation can be easily converted into message passing steps in the graph neural network.  Pseudo-likelihood training makes some sense, but energy loss minimization doesn’t really make sense and has serious known issues.

On the other hand, the proposed inference algorithms does not have good justifications.  Why not use something standard, like belief propagation for inference again?  Our community has studied graphical models a lot in the last decade and we have better algorithms than the ones proposed in this paper.

Lastly, the experiments are done on some standard but small benchmarks, and my personal experience with these datasets are that it is very easy to overfit, and most of the effort will be put in to prevent overfitting.  Therefore more powerful models typically cannot be separated from overly simple models.  I personally don’t care a lot about the results reported on these datasets.  Besides, there are a lot of questions about the proposed model, but all we get from the experiment section are a few numbers on the benchmarks.  I expect studies about this model from more angles.  One more minor thing about the experiment results: the numbers for GraphSAGE are definitely wrong.

Overall I think this paper tackles a potentially interesting problem, but it isn’t yet enough to be published at ICLR due to its problems mentioned above.

---

> ### Author Response · Authors · 2018-11-26
> **response, discussion and revision**
>
> We appreciate your detailed comments very much. The following are the clarification of some questions mentioned in the review.
> 1. Motivation: Although deep neural networks achieve good results in most prediction tasks, to our best knowledge, the output dependency has not yet been modeled in the representations. We can easily take two examples into consideration: (1). RNN-CRF used for sequence tagging and dense-CRF used for image segmentation. These models add CRF layer on top of a deep neural architecture in order to capture the tag dependency, and both achieved the state-of-the-art in their tasks. Obviously the output dependency is not modeled in the original deep representation, so adding the CRF layer could largely improve the results. (2). Adding a regularization term for labels improves the original deep neural networks, e.g. “Deep CNN with Graph Laplacian Regularization for Multi-label Image Annotation (Mojoo et al. 2017)”, and “Regularizing Prediction Entropy Enhances Deep Learning with Limited Data (Dubey et al. 2017)”. These examples demonstrate the usefulness of including output dependency in a deep neural network. Similarly for graph neural network, we believe that the output dependency could be useful as well.
>
> 2. Inference: We have removed the energy loss minimization section in the revised paper. One motivation of using pseudo-likelihood approximation (or even EBM loss) is to derive an exact form to approximate the objective function, thus making the back-propagation easily conducted in an end-to-end framework. Mean-field may be an alternative inference method, we will consider to adapt it to the graph structure and apply it to our model in future work.
>
> 3. As to the benchmark data, it may be true that “most of the effort will be put in to prevent overfitting”. In fact, the pairwise energy in our model can also be seen as a regularization, which prevents overfitting to some extent.
>
> 4. Experiments: We tried a new pytorch implementation of GraphSAGE (https://github.com/williamleif/graphsage-simple) on our datasets and the accuracies seem to become normal. The new results are updated in the paper. As to “a lot of questions about the proposed model” and “studies about this model from more angles”, could you elaborate some points?

---

> > ### Comment · AnonReviewer2 · 2018-12-03
> > **Thanks for the revision**
> >
> > Thanks for the revision.  But my main concerns (weakness in the learning algorithm and experiment results) are not addressed.
> >
> > w.r.t. motivation: I can fully understand why combining CRF and graph neural nets might be helpful, and indeed such combinations have been explored before in other domains like sequence and image pixel labeling.  However as far as I know at least for image segmentation, if you look at popular benchmarks like CityScapes or MS-COCO, the top-of-leaderboard entries are no longer using CRFs as the key component in the model.  This is largely due to the fact that the deep neural net can already model the dependency across input through the representations.
> >
> > w.r.t. the question: As to “a lot of questions about the proposed model” and “studies about this model from more angles”, could you elaborate some points?
> >
> > What I meant was, whenever you propose a new model with many new parts, we want to know as much information about it as possible from every aspect.  Currently all we know about the model comes from the 7 numbers in Table 3.  That's not enough to understand the properties of the model.  Admittedly understanding deep models is hard but we can do better.

---

### Official Review · AnonReviewer3 · 2018-11-05
**Interesting idea but not solid enough**

**Rating:** 4
**Confidence:** 5

**Review:**

The authors combine graph convolutional neural network and conditional random fields to get a new model, named conditional graph neural fields. The idea of the paper is interesting, but the work is not solid enough. Detailed comments come as follows,

1. In the proposed model, the authors treat the pairwise energy as prior and they do not depend on any features. Unlike the usual graph model in Eq (4), the authors further use the normalized \hat A as a scale factor in the pairwise term. What is the intuition for this?

2. The loss (10), which the authors claim that they are using, may not work. In fact, the loss cannot be used to use for training most architectures: ``while this loss will push down on the energy of the desired answer, it will not pull up on any other energy.''(LeCun et. al. 2006, A tutorial on energy-based learning). For deep structured model learning, please using piecewise learning, or joint training using some common CRF loss, such as log-loss. In fact, the authors are not using the energy-based loss as they have constraints on unary and pairwise terms. In fact, if we ignore the pairwise term in (11), the loss becomes log-loss for GCN. With the pairwise term, the loss is somehow like the loss for piecewise learning but the constraints on U is wrong (for piecewise learning, U should sum to 1).

3. The inference procedure is too simple that it can hardly find the near-optimal solutions. In fact, there exists an efficient mean-field based inference algorithm (Shuai Zheng et. al., 2015). Why did the authors choose a simple but poor inference procedure?

Comments After rebuttal
==========
Thank you for adress my concerns.

The response and the revision resolved my concern (1). However, the most important part, the possibly problematic loss is not resolved. It is true that sometimes (10) can achieve good results with good regularizers or a good set of hyperparameters. However, theoretically, the loss is ]only pushed down the desired answer, which may make the training procedure quite unstable. Thus I still think that a different loss should be used here.

---

> ### Author Response · Authors · 2018-11-26
> **response, discussion and revision**
>
> Thank you for the review and useful suggestions. Please find our response as follows:
> 1. Why normalized A:
> Notice that the pairwise energy is only calculated for two neighbor nodes, so the diagonal element of \hat{A} is never used. Thus it has the same effect as using the original adjacency matrix (except that the learned weight matrix U will have a different scale). Using \hat{A} instead of A is just for computational convenience.
>
> 2. About the energy loss (10):
> Yes, we admit the loss (10) may not work but it is not necessarily incorrect. In practice sometimes they can still achieve good results, as we showed in our experiments. Moreover, to avoid the problem of using loss (10), we removed the section of EBM based optimization in the revised paper.
>
> 3. The reason we did not choose the mean-field based inference algorithm in (Shuai Zheng et. al., 2015) was that it is originally used for CNN and the CRF inference is indeed transformed into a CNN module. But it cannot directly apply to our case, which has a different energy function and different data structures. For example, the message passing scheme cannot be treated as a convolution layer anymore. Using pseudo-likelihood approximation (or even EBM loss) could give us an exact form to approximate the objective function, thus easily making the framework end-to-end. But we do admit that the mean field inference may be adapted to the graph neural network and we will explore it in future work.

---

### Official Review · AnonReviewer1 · 2018-11-05
**Novelty is incremental.**

**Rating:** 5
**Confidence:** 4

**Review:**

This paper proposes a conditional graph neural network to explore the label correlation. In fact, this method combines graph convolutional neural network and CRF together. The novelty is incremental. No new insight is provided toward this topic.

For experiments, how did you train DeepWalk (Node2vec)? By using all nodes or the selected training set? It should be clarified in the paper.

Additionally, Table 3 says the result of semi-supervised methods. But how did you conduct semi-supervised learning for DeepWalk or Node2vec?

===================
After feedback:
Thanks for authors' feedback. Some of my concerns have been addressed. But the novelty is still not significant. On the other hand, the dataset used in this paper is simple. Specifically, at least the first 3 datasets are single-label and the number of classes is not large. They are too simple to support the claim. It's better to use multi-label datasets to show that the proposed method can really capture the correlation of labels.

---

> ### Author Response · Authors · 2018-11-26
> **response, discussion and revision**
>
> Thank you for the reviews. Here are the clarification about the novelty and experimental part:
>
> Insights: As we have claimed in the paper, our work have novelties from two perspectives.
> (1) From the perspective of node classification, we improve the graph neural networks by  consideration the label compatibility, which is a non-trivial problem. And combing GCN and CRF is the first work in this area.  (2) From the perspective of neural network structure, we admit that adding CRF to GNN is similar to adding CRF to other deep neural networks, such as RNN-CRF for sequence tagging and dense-CRF for image segmentation. However, how to formulate the energy function, and how to optimize the parameters are quite different in each model. The inference method for a sequence model or a CNN based model cannot be directly applied to our case.
>
> Experiments abou DeepWalk: For semi-supervised learning, we derive the embeddings of DeepWalk and Node2vec on all nodes and then train the node label classification using a one-layer MLP on the training data. This is almost identical to what is done for the semi-supervised graph neural networks. We have modified the paper and clarified this part.

---

### Meta-Review · Area_Chair1 · 2018-12-17
**Simple loss/inference, lack of thorough evaluation**

**Confidence:** 5
**Recommendation:** Reject

**Metareview:**


This paper introduces conditional graph neural fields, an approach that combines label compatibility scoring of conditional random fields with deep neural representations of nodes provided by graph convolutional networks. The intuition behind the proposed work is promising, and the results are strong.

The reviewers and the AC note the following as the primary concerns of the paper: (1) The novelty of this work is limited, since a number of approaches have recently combined CRFs and neural networks, and it is unclear whether the application of those ideas to GCNs is sufficiently interesting, (2) the losses, especially EBM, and the use of greedy/beam-search inference was found to be quite simple, especially given these have been studied extensively in the literature, and (3) analysis and adequate discussion of the results is missing (only a single table of numbers is provided).
Amongst other concerns, the reviewers identified issues with writing quality, lack of clear motivation for CRFs, and the selection of the benchmarks.

Given the feedback, the authors responded with comments, and a revision that removes the use of EBM loss from the paper, which the reviewers appreciated. However, most of the concerns remain unaddressed. Reviewer 2 maintains that CRFs+NNs still need to be motivated better, since hidden representations already take the neighborhood into account, as demonstrated by the fact that CRF+NNs are not state-of-art in other applications. Reviewer 2 also points out the lack of a detailed analysis of the results. Reviewer 2 focuses on the simplicity of the loss and inference algorithms, which is also echoed by reviewer 2 and reviewer 1. Finally, reviewer 1 also notes that the datasets are quite simple, and not ideal evaluation for label consistency given most of them are single-label (and thus need only few transition probabilities).

Based on this discussion, the reviewers and the AC agree that the paper is not ready for acceptance.